# Hybrid Composite Membrane of Phosphorylated Chitosan/Poly (Vinyl Alcohol)/Silica as a Proton Exchange Membrane

**DOI:** 10.3390/membranes11090675

**Published:** 2021-08-31

**Authors:** Nur Adiera Hanna Rosli, Kee Shyuan Loh, Wai Yin Wong, Tian Khoon Lee, Azizan Ahmad

**Affiliations:** 1Fuel Cell Institute, Universiti Kebangsaan Malaysia, Bangi 43600, Selangor, Malaysia; adierahanna@gmail.com (N.A.H.R.); waiyin.wong@ukm.edu.my (W.Y.W.); 2Faculty of Science and Technology, Universiti Kebangsaan Malaysia, Bangi 43600, Selangor, Malaysia; edison_tiankhoon@hotmail.com (T.K.L.); azizan@ukm.edu.my (A.A.)

**Keywords:** polymer blending, *N*-methylene phosphonic chitosan, poly (vinyl alcohol), silicon dioxide filler, proton exchange membranes

## Abstract

Chitosan is one of the natural biopolymers that has been studied as an alternative material to replace Nafion membranes as proton change membranes. Nevertheless, unmodified chitosan membranes have limitations including low proton conductivity and mechanical stability. The aim of this work is to study the effect of modifying chitosan through polymer blending with different compositions and the addition of inorganic filler on the microstructure and physical properties of *N*-methylene phosphonic chitosan/poly (vinyl alcohol) (NMPC/PVA) composite membranes. In this work, the NMPC biopolymer and PVA polymer are used as host polymers to produce NMPC/PVA composite membranes with different compositions (30–70% NMPC content). Increasing NMPC content in the membranes increases their proton conductivity, and as NMPC/PVA-50 composite membrane demonstrates the highest conductivity (8.76 × 10^−5^ S cm^−1^ at room temperature), it is chosen to be the base membrane for modification by adding hygroscopic silicon dioxide (SiO_2_) filler into its membrane matrix. The loading of SiO_2_ filler is varied (0.5–10 wt.%) to study the influence of filler concentration on temperature-dependent proton conductivity of membranes. NMPC/PVA-SiO_2_ (4 wt.%) exhibits the highest proton conductivity of 5.08 × 10^−4^ S cm^−1^ at 100 °C. In conclusion, the study shows that chitosan can be modified to produce proton exchange membranes that demonstrate enhanced properties and performance with the addition of PVA and SiO_2_.

## 1. Introduction

Nafion membranes have been commercially used as proton exchange membranes (PEMs) due to their high proton conductivity under hydrated conditions and good thermal and chemical stability. Nonetheless, aside from these advantages, Nafion membranes still have several shortcomings, including the high-cost of its materials, intense methanol crossover, and a severely decreased proton conductivity under low humidity conditions [1]. Moreover, the operation of PEMFC with the use of Nafion membrane was limited to low operating temperature (≈80 °C) [2,3]. Permanent hydration and gas humidification were needed to ensure high proton conductivity [4]. However, as the temperature exceeded 100 °C, the affinity with water and mechanical stability of Nafion membrane will be reduced [3]. Hence, various studies have been carried out over the years testing various biopolymer materials as alternative membrane materials to replace Nafion membranes.

Chitosan is a biopolymer material that has been considered an alternative material due to its hydrophilicity, biodegradability, biocompatibility, and low-cost. Additionally, due to the presence of free amine and hydroxyl groups in its backbone, chitosan can be chemically modified to produce functionalized chitosan composite membranes. Nonetheless, there are limitations that pristine chitosan membranes suffer from, namely, their comparatively poor mechanical properties and low proton conductivity (~10^−9^ S cm^−1^) under dry conditions and at room temperature because a small number of protons dissociated by moisture from the air in the chitosan matrix cannot move freely [5,6]. Numerous studies have been conducted and reported regarding chitosan modifications for developing enhanced chitosan membranes; these modifications include the addition of inorganic fillers, sulfonation, phosphorylation, quaternization, chemical cross-linking methods and blending with other polymers. One of the most popular modification methods that has been used over the past decade is the phosphorylation method. Wan et al. and Jayakumar et al. [7,8] reported that phosphorylated chitosan membranes could be prepared through the reaction of orthophosphoric acid and urea in *N,N*-dimethylformamide, as urea could promote the reaction, in which the produced functionalized membrane exhibited a decrease in its crystalline structure due to the increase in phosphorus content. In addition, *N*-methylene phosphonic chitosan (NMPC) was produced by Ramos et al., Binsu et al., Saxena et al., and Datta et al. [9,10,11,12] through a reflux method, whereas Dadhich et al. [13] produced NMPC by using a microwave-assisted rapid synthesis through a Mannich-type reaction. The produced NMPC had enhanced ionogenic and solubility properties without altering its filmogenic properties, allowing NMPC to be selected as a PEM for use in fuel cell applications [10,12,14].

The incorporation of inorganic materials into polymers has been extensively studied by past researchers. The addition of hygroscopic fillers, such as silicon dioxide (SiO_2_), titanium dioxide (TiO_2_) [15,16,17], and tungsten trioxide (WO_3_) [18,19], into the PEM matrix can affect the physicochemical properties of host matrix, including in improving mechanical properties [18], increasing the water retention capacity and proton conductivity of polymer composites by forming alternative proton conduction pathways [15,16,17]. Vijayalekshmi and Khastgir [20] studied and produced a series of chitosan-based nanocomposites with the addition of sulfonated polyaniline/nanosilica (sPAni/SiO_2_) to be used as proton exchange membranes. This CS-sPAni/SiO_2_ nanocomposite membrane showed a high protons conductivity of 8.39 × 10^−3^ S cm^−1^; additionally, the presence of SiO_2_ as an inorganic filler in the membrane enhanced the water uptake, improved proton transport, and provided additional pathways for proton conduction, which improved the proton conductivity [20]. Other than that, graphene oxide (GO), which is an active nanofiller [21,22] and could be modified with various functional groups [23,24], have been used in recent studies, in which the developed sulfonated graphene oxide (SGO) could enhance the proton conduction of polymeric membranes as well as provide continuous pathway for facile proton transport in the membranes [25,26]. Similarly, Bai et al. [27] synthesized phosphorylated graphene oxide (PGO), by allowing a polymeric layer bearing phosphonic acid (PA) groups as proton carriers covered onto GO surface, which contributed to the formation of efficient proton transfer channels along the membranes and achieved desired proton conductivity. The resultant membranes exhibited enhanced proton conductivity, thermal and mechanical stability [27].

Another method to improve the physical properties of chitosan membranes was through polymer blending with synthetic and natural polymer membranes. Several synthetic polymers, such as poly(vinyl alcohol) (PVA), poly(vinylidene fluoride) (PVDF), polyethersulfone (PES), polysulfone (PS), poly(ethylene oxide), polycaprolactone, and polyacrylamide, have been blended with chitosan to form composite membranes in previous studies. Blended membranes that undergo a cross-linking process demonstrate further improvements to their mechanical properties and water retention capacity [28]. The most common candidate that has been widely used to combine with chitosan membranes is PVA due to its unique properties, including its high crystallinity, water solubility, good film-forming ability, and high hydrophilicity due to containing reactive functional groups of -OH and forming hydrogen bonds that allowed chemical modification [29,30,31]. There have been several previous studies regarding the preparation of PVA/chitosan-blended membranes for fuel cell applications. PVA/chitosan and quaternized chitosan/quaternized PVA blended membranes have been prepared and cross-linked, exhibiting enhanced mechanical stability, methanol permeability, and proton conductivity and showing their potential use in direct methanol, anion exchange membranes and alkaline direct methanol fuel cell applications [32,33,34]. El Miri et al. [35] reported on the preparation and characterization of PVA/chitosan polymeric blends with the addition of cellulose nanocrystals, which act as nanoreinforcing agents, and this well-mixed membrane exhibited improved properties, including mechanical and thermal stability.

This study focuses on developing composite membranes containing the NMPC biopolymer and PVA polymer as the main hosts. These two polymers are mixed to form homogeneous solutions, which are then cast through the solution casting method and cross-linked to form NMPC/PVA composite membranes. The NMPC biopolymer is synthesized through the reflux method, while the used PVA polymer is commercially produced. The NMPC/PVA composite membranes are prepared with different compositions by varying the polymer ratios. A series of NMPC/PVA composite membranes modified with different loadings of SiO_2_ are prepared through the same solution casting method. The structural properties and the thermal and mechanical stability of these composite membranes are characterized by FTIR, XRD, FESEM, TGA, and DMA techniques. These membranes are also studied for their water uptake, swelling ratio, ion-exchange capacity, and proton conductivity to investigate the effect of polymer blending as well as the addition of SiO_2_ filler in the membranes at the low operating temperature (80–100 °C). The results of this study demonstrate that modifying chitosan through polymer blending and the addition of an inorganic filler, SiO_2_, produces composite membranes with enhanced performance when compared with the pure, unmodified NMPC membrane.

## 2. Materials and Methods

### 2.1. Materials

Commercial chitosan with a low molecular weight (50,000–190,000 Da, 75–85% deacetylated), phosphorous acid (99%), poly(vinyl alcohol) (Mw: 85,000–124,000, 99+% hydrolyzed), and silicon dioxide nanopowder (10–20 nm particle size (BET), 99.5%) were purchased from Sigma-Aldrich. Glacial acetic acid (99%), formaldehyde (37–40%), acetone (AR grade), sodium sulfate anhydrous (AR grade), and hydrochloric acid (37%, AR grade) were supplied by Systerm (Shah Alam, Selangor, Malaysia). Sodium hydroxide, sodium chloride, and phenolphthalein were obtained from R&M Chemicals (Petaling Jaya, Selangor, Malaysia). Sulfuric acid (95–97%) was procured from Merck (Kenilworth, NJ, USA). All materials were used without further purification. Deionized water was used throughout the whole experiment.

### 2.2. Synthesis of N-Methylene Phosphonic Chitosan (NMPC)

*N*-Methylene phosphonic chitosan (NMPC) was synthesized according to a previously reported phosphorylation method [9,10]. A chitosan solution was prepared by dissolving 5 g of chitosan powder in 250 mL of 1% (*v*/*v*) glacial acetic acid. Then, the solution was poured into a three-necked round-bottom flask with a magnetic stirrer and thermometer as well as a reflux condenser. The solution was then refluxed and heated with continuous stirring. The solution was heated continuously until the temperature reached 60 °C. Next, 2.5 g of phosphorous acid was dissolved in 25 mL of deionized water before being slowly added to the above solution. The heating process was continued until the temperature reached 70 °C, and 2.5 mL of formaldehyde was gradually added into the solution. The temperature of the solution was maintained for 8 h at 70 °C. The produced pale, yellow solution was then cooled to room temperature overnight. Acetone was excessively added into the solution until a white polymer precipitate was formed, which was the NMPC polymer. The resulting precipitate was filtered and washed with acetone in a Soxhlet apparatus for 24 h to remove unreacted phosphorous acid. Finally, the precipitate was dried in a desiccator.

### 2.3. Preparation of the N-Methylene Phosphonic Chitosan/Poly (Vinyl Alcohol) Composite Membrane

The *N*-methylene phosphonic chitosan/poly (vinyl alcohol) (NMPC/PVA) composite membranes were prepared by the solution casting method [10]. NMPC and PVA solutions were prepared by separately dissolving their respective compounds in known amounts of water before producing various compositions of NMPC/PVA composite membranes. Then, both solutions were mixed dropwise under continuous stirring for 5 h at room temperature. Air bubbles that formed in the solution were removed by sonication to obtain a clear solution, which was then cast onto a clean glass petri dish and dried for 4 days at 60 °C. The resulting dried membranes (films) were further subjected to cross-linking through immersion in a solution containing formaldehyde (54.1 g), sodium sulfate (150.0 g), sulfuric acid (125.0 g), and water (470.0 g) for 2 h at 60 °C. Next, the cross-linked films were washed with deionized water to remove the unreacted cross-linking agents and were further dried at ambient temperature for 24 h. The obtained membranes were designated NMPC/PVA-X, where X was the NMPC content (%, w/w; 30–70%) in the membrane phase. The content range of NMPC/PVA membranes was determined based on the preliminary studies that have been conducted. The membrane samples were kept in a desiccator to avoid exposure to moisture before further characterization.

### 2.4. Preparation of the N-Methylene Phosphonic Chitosan/Poly (Vinyl Alcohol) Composite Membrane Modified with Silicon Dioxide Filler (NMPC/PVA-SiO_2_)

NMPC/PVA-SiO_2_ composite membranes were prepared through the same solution casting method as the NMPC/PVA composite membrane. The effect of a hygroscopic material on the NMPC/PVA composite membrane was studied by adding SiO_2_ filler into the polymer solution. The NMPC/PVA-50 composite membrane with the highest proton conductivity value was selected as the base membrane to conduct this study. SiO_2_ powder was dispersed into the NMPC/PVA polymer solution at a ratio of 50:50 (%, *w*/*w*) using a sonication process in a water bath and stirred continuously with different SiO_2_ loadings (0.5, 1, 2, 4, 6, 8, and 10 wt.%) until becoming homogeneous. The basis for the selection of SiO_2_ loadings range was determined based on the range that was commonly used in the preliminary studies. The solution was then poured into a glass Petri dish and dried for 4 days at 60 °C, and all the resulting NMPC/PVA-SiO_2_ composite membranes underwent a cross-linking process before characterization was conducted.

### 2.5. Characterization

#### 2.5.1. Fourier Transform Infrared (FTIR) Spectroscopy

FTIR analysis was conducted using a Perkin Elmer Spectrum 400 FTIR/NIR spectrometer (Perkin Elmer, Ohio, USA) in attenuated total reflectance (ATR) mode in the wavenumber region of 4000–650 cm^−1^ with a scan resolution rate of 4 cm^−1^ at room temperature condition. FTIR studies were performed to determine the functional groups of the pristine chitosan and NMPC powder, as well as the produced composite membranes.

#### 2.5.2. X-ray Diffraction (XRD)

X-ray diffraction was performed using a Bruker D8 Advance diffractometer to determine the crystallinity of the pristine chitosan, NMPC powder, silicon dioxide nanopowder (commercially purchased), along with the resulting composite membranes. The diffractograms were measured with Cu-K_α_ radiation (wavelength of radiation = 0.15405 Å) at diffraction angles (2θ) in the range of 5–80°. The crystallinity and amorphous phases of the polymer and membrane samples were measured using Bruker Diffrac EVA XRD software (Bruker, Massachusetts, USA).

#### 2.5.3. Field-Emission Scanning Electron Microscopy (FESEM)

The cross-sectional morphologies of the NMPC membrane, NMPC/PVA and NMPC/PVA-SiO_2_ composite membranes were observed using field-emission scanning electron microscopy (FESEM, Zeiss SUPRA 55VP, Jena, Germany). The instrumentation was equipped with an element mapping energy-dispersive X-ray spectroscopy (EDX) analyzer to observe the homogeneity and distributions of elements on the modified composite membranes according to the predetermined filler compositions. Before the analysis was performed, all samples will be deposited with gold or carbon through vacuum spraying to form an ultra-thin flow layer without changing the morphological structure of the samples and the samples were analyzed at magnification level of 1000×.

#### 2.5.4. Thermogravimetric Analysis (TGA)

A thermogravimetric analyzer (Perkin Elmer STA 6000, Akron, OH, USA) was used to determine the thermal stability and thermal degradation process of the membranes in a nitrogen atmosphere at a heating rate of 10 °C min^−1^ from 30–600 °C.

#### 2.5.5. Dynamic Mechanical Analysis (DMA)

Mechanical strength analysis of the membranes was conducted under isothermal conditions using a dynamic mechanical analyzer (Perkin Elmer DMA 8000, Akron, OH, USA) at a frequency of 1 Hz and a heating rate of 3 °C min^−1^ from 25–200 °C.

#### 2.5.6. Water Uptake and Swelling Ratio

For the measurement of water uptake, the membrane samples were dried at room temperature for a few days, and their weights were measured continuously for 3 days to ensure that the membrane’s weight is constant. Then, the membranes were immersed in deionized water for 24 h at room temperature. The membranes were wiped off with tissue paper, and excess surface water was removed. The weight of the wet membranes was measured, and the weight differences before and after hydration in relation to the weight of the dry membranes were calculated as the water uptake percentage using Equation (1):(1)$$\mathrm{Water}\;\mathrm{uptake}\;\left(\%\right)=\frac{W_{\mathrm{wet}}-{\;W}_{\mathrm{dry}}}{W_{\mathrm{dry}}}\times 100$$
where W_wet_ is the weight of membranes after being immersed in deionized water and W_dry_ is the initial weight of dry membranes.

The swelling ratio of the membrane samples was measured by measuring the change in surface area and thickness before and after hydration. The thickness of the membranes was measured with a micrometer, and the swelling ratios were calculated from Equations (2) and (3), respectively:(2)$$\mathrm{Swelling}\;\mathrm{area}\;\left(\%\right)=\frac{A_{\mathrm{wet}}-{\;A}_{\mathrm{dry}}}{A_{\mathrm{dry}}}\times 100$$
(3)$$\mathrm{Swelling}\;\mathrm{thickness}\;\left(\%\right)=\frac{T_{\mathrm{wet}}-{\;T}_{\mathrm{dry}}}{T_{\mathrm{dry}}}\times 100$$
where A_wet_ and T_wet_ represent the surface area and thickness of the membranes after being immersed in deionized water, respectively. A_dry_ and T_dry_ are the initial surface area and thickness of dry membranes, respectively. The average value of swelling area and thickness was measured from three measurements.

#### 2.5.7. Ion-Exchange Capacity (IEC)

The ion-exchange capacity (IEC) of the membrane samples was determined through the usual acid-base titration method. The membranes were equilibrated in 1.0 M HCl solution for 24 h to convert the membrane into the H^+^ form. The membranes were then washed with deionized water to remove excess HCl. Next, the membranes were equilibrated in 0.1 M NaCl solution for 24 h and titrated against a 0.1 M NaOH solution by using phenolphthalein as the universal indicator. The IEC (mequiv g^−1^) values were calculated using Equation (4):(4)$$\mathrm{IEC}=\frac{V_{\mathrm{NaOH}}\times {\;C}_{\mathrm{NaOH}}}{W_{\mathrm{dry}}}$$
where V_NaOH_ is the volume of NaOH used, C_NaOH_ is the concentration of NaOH and W_dry_ is the initial weight of dry membranes.

#### 2.5.8. Proton Conductivity

The proton conductivity of the membrane samples was measured at room temperature for NMPC membrane and NMPC/PVA composite membranes, whereas the conductivity of NMPC/PVA-SiO_2_ composite membranes was measured at the temperature of 25–100 °C, under hydrated conditions. The proton conductivity was measured by electrochemical impedance spectroscopy (EIS) using an electrochemical workstation (Autolab PGSTAT 128N, Utrecht, The Netherlands) and a signal amplitude of 10 mV over a frequency range of 1 Hz to 1 MHz. The membrane samples were immersed in deionized water for 24 h at room temperature and then placed into a clamp and connected by two platinum electrodes. The measurements were made by placing a membrane disc with a diameter of 2 cm^2^ into a Teflon conductivity closed cell containing two stainless steel electrodes in a temperature-controlled chamber. A little amount of water in the cell maintained the relative humidity at 100%. The proton conductivity was measured from room temperature, 25 °C to 100 °C, and the samples were kept at each temperature for 15 min for the membrane to reach an equilibrium temperature [36,37]. The proton conductivity was determined from a Nyquist plot to obtain the bulk resistance of the membrane through the intersection of the high-frequency intercept with the real axis. The proton conductivity values were calculated using Equation (5):(5)$$\sigma =\frac{t}{R_{b}\times A}$$
where t and A are the thickness and area of the prepared membrane samples, respectively, and R_b_ is the bulk resistance of the membrane samples. The proton conductivity values were the average of three measurements, and the standard deviation was calculated.

## 3. Results and Discussion

### 3.1. Chitosan and N-Methylene Phosphonic Chitosan (NMPC)

#### 3.1.1. FTIR and XRD Analysis

The FTIR spectra of chitosan and NMPC are presented in Figure 1a (i and ii), which was conducted to determine and differentiate the chemical structures and bonds between both derivatives. The FTIR spectra of pristine chitosan (Appendix A) showed a broad peak from 3500–3200 cm^−1^, which represented the overlapping O–H and N–H stretching bands [38]. The absorption peak at 2871 cm^−1^ (Appendix A) was assigned to C–H stretching, whereas the weak absorption peak at 1643 cm^−1^ corresponded to C=O stretching due to amide carboxyl groups; additionally, the peak at 1584 cm^−1^ represented N–H amine bending (Appendix A). The peaks at 1422, 1376, and 1311 cm^−1^ (Appendix A) were ascribed to C–N stretching coupled with N-H in-plane deformation, symmetric angular –CH_3_ bending, and C–N stretching of the amino group, respectively [38,39]. The peak at 1149 cm^−1^ indicated C–O–C stretching, whereas the strong bands at 1065 cm^−1^ and 1026 cm^−1^ were attributed to the presence of C–O stretching in the chitosan skeleton (Appendix A) [12,38].

The FTIR spectra also represented the chemical bonds that existed in NMPC (Figure 1a (ii)), resulting from the phosphorylation of chitosan. The NMPC spectra showed the O–H and N–H stretching bands at 3500–3200 cm^−1^ broadened, implying the substitution of –CH_2_PO_3_H_2_ by the H atoms in the amine groups and affecting the hydrogen bonds [12,40]. The amine deformation peaks shifted to lower frequencies, from 1643 cm^−1^ and 1548 cm^−1^ in chitosan to 1632 cm^−1^ (antisymmetric deformation) and 1536 cm^−1^ (symmetric deformation) in NMPC; this result indicated the protonation of the chitosan amine as there was a hydrogen substitution to the methylene phosphonic groups that made it a tertiary amine and involved both peaks attributed to NH_3_^+^ groups [12,39]. The new bands at 1243 cm^−1^ and 943 cm^−1^ were ascribed to P=O and P-OH stretching bands, whereas the peaks at 1470 cm^−1^ and 1380 cm^−1^ were assigned to –CH_2_– vibrations of the methylene phosphonic groups in the molecule [12,39,41]. Moreover, the new peak at 2386 cm^−1^ indicated the P–H stretching of phosphonic groups, and the existence of these new peaks proved the addition of methylene phosphonic groups into chitosan. In addition, the strong bands with high intensities at 1057 cm^−1^ and 1027 cm^−1^ implied that C–O stretching overlapped with P–OH stretching; thus, according to the FTIR analysis, the resultant NMPC was successfully synthesized through a phosphorylation method [39,41].

The phase composition and structure of pristine chitosan and NMPC were analyzed using XRD. The XRD diffraction pattern of pristine chitosan (Figure 1b (i)) showed that chitosan had semicrystalline properties, consisting of both amorphous and crystalline phases. The existence of a sharp peak at 2θ = 19.90° and another peak at 2θ = 10.30° showed a high degree of crystallization in chitosan. The high degree of crystallization in chitosan was due to intra- and extra-molecular hydrogen bonding [42].

The XRD spectrum of NMPC (Figure 1b (ii)) showed reflections at 2θ values of 11.40° and 21.80°, which were attributed to amorphous properties; additionally, these values were far from those found in chitosan and were representative of a novel structure that accommodated the bulky substituents. NMPC, which is a water-soluble derivative, has a high degree of amorphous phase and can reduce the crystallinity of chitosan; thus, the movement of the polymer chain segment can be enhanced and improve the proton conductivity values [43].

### 3.2. N-Methylene Phosphonic Chitosan/Poly (Vinyl Alcohol) (NMPC/PVA) Composite Membranes

#### 3.2.1. FTIR and XRD Analysis

FTIR analyses were performed to identify the chemical structure of unmodified NMPC and PVA membranes and prepared NMPC/PVA composite membranes. Figure 2a,b shows the FTIR spectra of the NMPC and PVA membranes along with the NMPC/PVA-50 composite membrane. In Figure 2a (i), regarding the NMPC membrane, the representative bands and peaks have been described and explained in Section 3.1.1. In addition, Figure 2a (ii) shows the FTIR spectrum of the PVA membrane, in which the broad and strong band between 3500 and 3100 cm^−1^ was attributed to O–H stretching from the intramolecular and intermolecular hydrogen bonds. The existence of peaks at 2942 cm^−1^ and 2910 cm^−1^ were attributed to the symmetric and antisymmetric stretching vibrations of C–H from alkyl groups. Moreover, the peaks at 1721 cm^−1^ and 1656 cm^−1^ were related to C=O stretching vibrations, while the peak at 1091 cm^−1^ was attributed to the C-O stretching band, and these stretches were assigned to the remaining acetate groups during the production of PVA molecules from the hydrolysis of polyvinyl acetate [44,45,46,47,48]. The peak at 1558 cm^−1^ represented the O-H bending vibration of hydroxyl groups [35], while the presence of sharp bands at 1415, 1331, and 844 cm^−1^ corresponded to -CH_2_, -CH_3_, and C-H bending, respectively.

The FTIR spectra of NMPC/PVA composite membranes with different compositions are shown in Figure 2a (iii) as well as in Figure 2b (i–ix). The broad band exhibited at 3405 cm^−1^ indicated N-H stretching, overlapping with O-H stretching from both the NMPC and PVA molecules (Appendix A). The intensity of this broad peak became weak when compared to unmodified NMPC and PVA membranes, which verified the formation of the NMPC/PVA composite membrane through hydrogen bonding between the NMPC segments and PVA chains (Figure 2a (iii)) [10,11]. The peaks at 2943, 2918, and 2862 cm^−1^ were assigned to C-H stretching (Appendix A), whereas the peak at 1644 cm^−1^ indicated N-H bending from the -NH_3_^+^ groups of NMPC and overlapped with the C=O stretching vibration (Appendix A). Additionally, with an increasing NMPC content in the polymer matrix, the intensity of the peak for the absorption band at 1644 cm^−1^ increased. Moreover, the presence of peaks at 1476, 1431, and 1394 cm^−1^ corresponded to -CH_2_ and -CH_3_ bending from the PVA segment and methylene of the NMPC molecule (Appendix A). The peaks at 1132 cm^−1^ and 1062 cm^−1^ and the sharp peak with high intensity at 1006 cm^−1^ were ascribed to the C-O stretching band, which overlapped with P-OH stretching, while the peaks at 838 cm^−1^ and 782 cm^−1^ signified C-H bending from both the PVA and NMPC molecules (Appendix A).

XRD analyses were performed to confirm the change in the degree of crystallinity of unmodified NMPC and PVA membranes, as well as NMPC/PVA composite membranes. Figure 2c illustrates the XRD spectra of the unmodified NMPC and PVA membranes. The peaks at 2θ = 11.40° and 21.80° showed that an amorphous structure existed in the NMPC polymer, with the amorphous percentage of 87.8% (Figure 2c (i)) [43]. Additionally, the PVA membrane showed a peak at 2θ = 19.50°, which corresponded to the (1 0 1) plane, resulted from the semi-crystalline region of PVA with crystalline percentage of 47.2% and in contrast, the peak at 2θ = 40.00° was a broad band that indicated the amorphous region in the PVA polymer (Figure 2c (ii)) [49,50].

The peaks at 2θ = 19.50° for all NMPC/PVA composite membranes with different compositions exhibited the presence of amorphous regions in each membrane (Figure 2d (i–ix)). The presence of a low intensity peak at 2θ = 29.50° of the NMPC/PVA-45 composite membrane was likely due to the presence of impurities in the resulting membrane sample during the process of membrane production. XRD analysis showed that the crystallinity of the membranes decreased with an increasing NMPC content; hence, the amorphous region increased. The amorphous percentage of NMPC/PVA composite membranes increased with NMPC content of 30–50%, with a percentage increase of 50.6–68.7%, correlated with the increase of proton conductivity, while when the NMPC content increased up to 70%, the amorphous percentage decreased to 55.7%. The decrease in the crystallinity of membranes was due to the intermolecular interaction between both polymers that could destroy the hydrogen bonding between polymer chains and suppress the crystallinity. The decrease in crystallinity phase of membrane caused an increase in the movement of polymer chain segments, in turn aiding in the improvement of proton conductivity.

#### 3.2.2. Morphological Studies

Appendix A displayed the digital images of NMPC membrane and NMPC/PVA-50 composite membrane prepared in this work. Images were captured after drying the NMPC membrane and after cross-linking and drying process of the NMPC/PVA-50 composite membrane. Both images of NMPC-based membranes were yellowish in color (Appendix A), whereas the NMPC/PVA-50 composite membrane exhibited an opaque and less flexible membrane with the addition of PVA (Appendix A). Moreover, through physical observation and measurement of sample thickness using a thickness gauge, NMPC membrane has a lower thickness (0.05 mm) compared to NMPC/PVA composite membrane which has a thickness of around 0.10 mm.

Changes in the morphology of the NMPC membrane, PVA membrane and NMPC/PVA composite membranes could be observed using FESEM (Figure 3). The FESEM micrographs in Figure 3 showed cross-sectional views of the NMPC membrane, PVA membrane, and NMPC/PVA composite membranes with different compositions, and the unmodified NMPC membrane displayed a dense, smooth, and homogeneous structure without obvious pores (Figure 3a). On the other hand, the PVA membrane displayed a porous and homogeneous structure (Figure 3b). Figure 3c–k show that all composite membranes were homogeneously combined, and the structure of the composite membranes appeared fibrous with shallow pores. As shown in Figure 3c–k, the cross-sectional morphology has become rough when the NMPC polymer combined with the PVA polymer. Nevertheless, no phase separation occurred when combining both polymers, thus proving that they were compatible with each other when producing NMPC/PVA composite membranes.

Based on Figure 3c–k, the structures of all NMPC/PVA composite membranes exhibited porous structures due to the presence of PVA, which has hydrophilic properties and excellent water permeability. The structure of a membrane plays an important role in controlling these factors because porous structures exhibit high flux or water permeability and low selectivity, while dense structures show the opposite. The NMPC/PVA composite membrane showed porous morphology due to the cross-linking process that occurred with all composite membranes [51].

#### 3.2.3. Thermal Stability

Thermogravimetric analysis (TGA) was carried out to determine the thermal stability of the NMPC membrane and NMPC/PVA composite membranes. The TGA curve of the NMPC membrane in Figure 4 showed two stages of weight loss. The first stage of weight loss was approximately 17% over the temperature of 70 °C, which was attributed to the loss of the moisture content in the membrane as well as the elimination of side groups. The second stage of weight loss occurred at approximately 200 °C with a weight loss of approximately 40% because the decomposition of functional groups existed in NMPC biopolymer (CH_2_, NH_2_, and PO_3_H).

In contrast, the TGA curves presented in Figure 4 show that all prepared NMPC/PVA composite membranes followed similar degradation behavior. There were three stages of degradation that occurred in the NMPC/PVA composite membranes as shown in Figure 4 (I, II, and III). The first stage of degradation occurred from approximately 80–100 °C, with a weight loss of approximately 9–10% that was due to the loss of absorbed water molecules in the membrane matrix. The second weight loss was in the range of 230–300 °C, which was ascribed to the loss of PVA polymer and the thermal degradation of the cross-linking network formed in the membrane matrix. The second stage weight loss occurred at around 230 °C indicated that there was an increase in the thermal stability of this modified biopolymer due to the cross-linking that occurred in the membrane [10]. The third stage of degradation was due to the decomposition of the main polymer chain in the membrane from approximately 370–460 °C with a weight loss of approximately 37–54%. Appendix A exhibited the thermal stability analysis of NMPC membrane and NMPC/PVA composite membranes with different compositions according to the three degradation stages and weight losses (I, II, and III), as labelled in Figure 4. The thermal stability of the NMPC/PVA composite membranes slightly improved when compared to the unmodified NMPC membrane because of the hydrogen bond interaction between the NMPC and PVA polymers. Throughout this study, it could be concluded that the NMPC/PVA-65 composite membrane showed the highest thermal stability among the other membranes. The improvement in the thermal stability of the NMPC/PVA-65 composite membrane was due to its higher residual content (approximately 25%) compared to the NMPC/PVA composite membrane with other ratios (13–22%). However, there are some factors that could be considered to affect the thermal stability of the NMPC/PVA-70 composite membrane, which demonstrated less residual content, including the crystallinity of the membrane and degree of cross-linking that has not been determined in this study.

#### 3.2.4. DMA Analysis

Dynamic mechanical analysis (DMA) is a sensitive technique that yields information on bulk properties and thermal transitions, as well as other minor phase or structural changes of polymers [52]. The dynamic mechanical properties were demonstrated by the tan δ, storage modulus, and loss modulus. The tan δ and loss modulus peaks were described as the glass transition temperature, where the tan δ peak occurred at a higher temperature than the loss modulus [53]. Tan δ is known as a good limit of the leather-like midpoint between the glassy and rubbery states, whereas the storage modulus is a limit of the recoverable stored strain energy, while the loss modulus is a limit of the energy consumed, which is lost as heat [52,54].

Figure 5a–i shows the changes in tan δ and storage modulus versus temperature for the NMPC/PVA composite membranes with different compositions, whereas Appendix A shows the comparisons of tan δ curves for all NMPC/PVA composite membranes. The sharp peaks presented on the tan δ curves represented T_g_ and the plot on the storage modulus was a transition corresponding to the presence of peak at tan δ. As the process was moving from a very low temperature, the molecules of the composite membranes were tightly compressed to higher temperatures. The molecules were expanded, and the free volume increased, allowing side chain movement to occur; this behavior is known as a beta transition (T_β_), which was often related to toughness. The glass transition (T_g_), which only occurred in the amorphous materials, appeared as the heating process continued, and the chains in the amorphous regions began to coordinate into large-scale chains; thus, the amorphous region started to melt into a rubbery phase [55,56]. Based on Figure 5a–i and Appendix A, on the tan δ curve, it can be observed that T_g_ occurred between 118 and 130 °C for all composite membranes. In addition, the range of T_g_ peak heights for all composite membranes was 0.3–0.6 and could be seen on the tan δ curve. This T_g_ value represented the amorphous phase in the membrane and could be zero for samples that were in fully crystalline phase [55]. Figure 5h–i shows that the T_g_ peak value on tan δ curve (around 0.3) for NMPC/PVA-65 and NMPC/PVA-70 composite membranes experienced a decrease in the amorphous phase, corresponded to the decrease in amorphous percentage discussed in the XRD analysis (Section 3.2.1). According to the graphs, another peak might exist at temperatures below 35 °C, although the data did not include sufficiently low temperatures to fully capture this peak [52].

The storage modulus for all NMPC/PVA composite membranes is shown in Figure 5a–i, in which the trend was quite inconsistent. The inconsistency of this trend might be contributed by the thickness and degree of cross-linking of the composite membranes. As the NMPC content increased to 35% (Figure 5b), the storage modulus increased (2.46 × 10^6^ Pa) and then decreased slightly as the NMPC content further increased. Furthermore, the storage modulus value decreased as 70% NMPC content (Figure 5i) was added to the membrane. The storage modulus values for the composite membranes with NMPC contents of 30–55% experienced a decrease when the temperature increased to around 100 °C. The NMPC/PVA composite membranes with 60–70% NMPC content (Figure 5g–i) exhibited a storage modulus peak or hump directly at the edge of the preceding drop that correlated to the T_g_. This peak or hump referred to the overshoot at T_g_, which was caused by stress relief. The stress was trapped in the membrane matrix until enough movement was attained at the T_g_ to induce the movement of chains to a lower energy state [57]. Additionally, the storage modulus of the initial value (at a temperature of around 35 °C) was maintained at approximately 100 °C in the NMPC/PVA composite membranes with NMPC content of 60–70% (Figure 5g–i). The retention of these storage modulus values demonstrated the mechanical stability of the composite membrane at higher temperature, making it potentially applicable for fuel cell applications.

#### 3.2.5. Water Uptake, Swelling Ratio, Ion Exchange Capacity, and Proton Conductivity

The proton conductivity of membranes was closely related to the water uptake values through the hydration degree. Proton conductivity could be promoted as hydrogen bond networks formed between the membrane and water molecules under hydrated conditions. In this study, the water uptake, swelling area, and swelling thickness of composite membranes at room temperature were measured and are presented in Table 1. As expected, pure, unmodified NMPC had a high degree of hydrophilicity, as it dissolved immediately when it was immersed in water or acetic acid; hence, its water uptake and swelling ratio could not be measured. Regarding the NMPC/PVA composite membranes, the water uptake increased with an increasing NMPC content in the membrane matrix (32.1%–51.9%) and then slightly decreased at some point. The highest water uptake (51.9%) was observed in the same ratio content of NMPC and PVA (NMPC/PVA-50). The reason for this increase was due to the presence of hydrophilic groups (–NH_2_ and –OH) in NMPC and PVA, which helped to attract water.

The swelling ratios, which were the swelling area and swelling thickness of all NMPC/PVA composite membranes, are also shown in Table 1. Generally, a higher water uptake can induce a higher swelling ratio. Table 1 shows that both the swelling area and swelling thickness increased with increasing water uptake values and slightly decreased when the percentage of water uptake decreased; all the composite membranes exhibited similar trends. Additionally, all NMPC/PVA composite membranes underwent a cross-linking process. Ionic cross-linking might not be sufficient to depress the swelling of composite membranes; however, the covalent cross-linking that occurred in the membrane matrix could inhibit the dissolution and excess swelling of the composite membranes [10].

The IEC represented the number of active sites or functional groups that were responsible for ion exchange in the polymer electrolyte membranes and was also related to proton conduction for the composite membranes. The IEC values for the NMPC/PVA composite membranes with different compositions are presented in Table 1. The increase in NMPC content (30–50%) in the NMPC/PVA composite membranes increased the IEC magnitude from 0.24 to 0.45 mequiv g^−1^ and decreased (0.38–0.29 mequiv g^−1^) to some extent when the NMPC content was further increased (55–70%). Based on the IEC values shown, it can be concluded that the increase in density of -PO_3_H_2_ groups contributed to the improvement in IEC, which was predicted to enhance the proton conductivity of the NMPC/PVA composite membranes [10]. However, as the content of NMPC exceeded 50 wt.%, the IEC values experienced decrement, which might be due to the degree of cross-linking of membranes that has not been determined and the decrease in percentage of amorphous region, hence the conductivity values were also expected to be decreased.

Proton conductivity is a significant feature of fuel cell membranes, as the efficiency of fuel cell performance depends on proton conductivity. Appendix A shows the typical Nyquist plots, with semicircles at lower frequencies and the linear diffusion impedance on the x-axis, which represented the bulk resistance (R_b_) of the membrane matrix. The proton conductivity of the NMPC/PVA composite membranes with different compositions was measured at room temperature under hydrated conditions and is shown in Table 1. The proton conductivity of the pure, unmodified NMPC membrane was measured under dry condition, as it could not be immersed in water due to its dissolution effect and low proton conductivity (2.74 × 10^−6^ S cm^−1^). Additionally, in the NMPC/PVA composite membranes, the proton conductivity increased with an increasing NMPC content in the membrane matrix from 30–50%, with values of 2.61 × 10^−5^–8.76 × 10^−5^ S cm^−1^, respectively, and then decreased as the NMPC content was further increased. The increase in proton conductivity of the composite membranes might be due to the presence of zwitterionic architecture (Figure 6), which gave rise to hydrophilic regions in the membrane matrix because of its strong affinity toward water. These hydrophilic regions promoted the absorption of water and facilitated proton transport, and with the increase in NMPC content in the membrane matrix, protons existed in the form of H_3_O^+^ passing through the hydrophilic regions and increasing the proton conductivity [10]. This increase in proton conductivity was correlated with the increase in water uptake and the IEC, which synchronized with the increase in hydrophilic groups in the NMPC/PVA composite membranes. Proton transport in the membrane is known and can be described by two mechanisms: vehicle and Grotthuss (hopping) mechanisms. The bound water in the membrane was probably associated with the Grotthuss mechanism, while the free water in the membrane participated in the vehicle mechanism, proving that the presence of water was vital for proton conduction in the membrane [58,59].

### 3.3. N-Methylene Phosphonic Chitosan/Poly (Vinyl Alcohol) Composite Membranes Modified with SiO_2_ Filler (NMPC/PVA-SiO_2_)

#### 3.3.1. FTIR and XRD Analysis

FTIR analysis was one of the most important characterizations and was performed to identify the chemical structures present in the filler and produced composite membranes. Figure 7a shows the FTIR spectra for SiO_2_, the NMPC/PVA-50 composite membrane and the NMPC/PVA-SiO_2_ (4 wt.%) composite membrane. The FTIR spectra of the SiO_2_ filler (Figure 7a (i)) exhibited a band at 3348 cm^−1^, which was associated with the O-H stretching vibration of absorbed water molecules, and the peak at 1654 cm^−1^ was the bending vibration of these water molecules. The absorption band at 1057 cm^−1^ corresponded to the asymmetric stretching mode for the vibration of Si-O-Si bonds in the SiO_2_^4-^ four-coordinate species (Si-(SiO)^4^=Q^4^) in the oxide phase on the surface, which was a typical band for the pure and amorphous SiO_2_ fillers [60]. The broad band in the 3800–2500 cm^−1^ region represented hydrogen bonds from the different O-H groups present in SiO_2_. Moreover, the region at 1200–770 cm^−1^ showed that SiO_2_ had specific bands focused on the Si-O-Si, Si-O, and Si-OH stretching vibrations, and the peak at 796 cm^−1^ referred to the vibration mode of O-Si-O from SiO_2_ crystals [61,62,63].

The representative FTIR peaks and bands of NMPC/PVA membranes (Figure 7a (ii)) have been explained in Section 3.2.1. The FTIR spectra of the NMPC/PVA-SiO_2_ composite membranes are shown in Figure 7a (iii), as well as in Figure 7b (i–vii), with different compositions of SiO_2_ fillers. The NMPC/PVA-SiO_2_ composite membrane showed no significant change in the FTIR spectrum compared to the unmodified NMPC/PVA composite membrane, and the main vibration spectra presented were quite similar. The broad band exhibited at 3405 cm^−1^ showed N-H stretching, overlapping with the O-H stretching from the NMPC, PVA, and SiO_2_ molecules (Appendix A). This broad peak intensity became weak when compared to the NMPC/PVA composite membranes, confirming the formation of NMPC/PVA-SiO_2_ composite membranes through hydrogen bonding between the NMPC segments, PVA chains, and SiO_2_ segments. The peaks at 2943, 2915, and 2862 cm^−1^ were associated with C-H stretching (Appendix A). The peak at 1645 cm^−1^ shifted to a lower frequency of 1639 cm^−1^ and was attributed to the N-H bending from the -NH_3_^+^ group in NMPC, overlapping with the C=O stretching vibration and bending vibration of water molecules from the SiO_2_ segment (Appendix A). The peaks at 1476, 1432, and 1406 cm^−1^ corresponded to the -CH_2_ and -CH_3_ bending of the PVA segment and methylene phosphonic group in the NMPC molecule (Appendix A). The peaks at 1129 cm^−1^ and 1065 cm^−1^ and the sharp peak with high intensity at 1007 cm^−1^ were attributed to the C-O stretching bands, which overlapped with the P-OH stretching from the NMPC molecule and can be seen in Appendix A. The peak at 1065 cm^−1^ shifted to a lower frequency of 1055 cm^−1^ due to Si-O-Si stretching and Si-OH stretching at 1007 cm^−1^ of the SiO_2_ molecules in that region. In addition, the peaks at 834 cm^−1^ and 783 cm^−1^ indicated that the C-H bending from both the PVA and NMPC molecules overlapped with the Si-O-Si stretching and O-Si-O vibration of the SiO_2_ molecules (Appendix A).

XRD analysis was performed to demonstrate the crystallinity and confirm changes in the degree of crystallinity of the SiO_2_ filler and NMPC/PVA-SiO_2_ composite membranes. Figure 7c illustrated the XRD spectrum of the commercial SiO_2_ filler obtained experimentally, which exhibited a highly crystalline structure with crystalline percentage of 91.5%. Dominant crystalline peaks were observed at 2θ = 20.89° and 26.67°, representing SiO_2_ with a low type of cristobalite and a tetragonal shape (PDF Card No.: 01-076-0941) [64].

The NMPC/PVA-50 composite membrane showed the presence of an amorphous region at 2θ = 19.50° (Figure 7d (i)). Peaks at 2θ = 19.50° for all NMPC/PVA-SiO_2_ composite membranes with different compositions exhibited the presence of amorphous regions in each membrane and proved that both the NMPC/PVA polymer matrix and SiO_2_ filler were well dispersed (Figure 7d (ii–viii)). The presence of a diffraction peak at an angle of 2θ = 26.67° indicated the presence of SiO_2_ filler in the polymer matrix. The peak was not visible in the XRD spectrum with the addition of 0.5–2 wt.% SiO_2_ filler into the polymer matrix. This invisible peak was likely due to the small amount of SiO_2_ loading; however, the peak was clearly visible when the SiO_2_ loading was increased up to 10 wt.%. The uniformity factor of filler distributions in the membrane was likely to play a role in the existence of uneven peaks in the XRD spectrum. In addition, it can be observed that with the addition of SiO_2_ loadings of 0.5–4 wt.%, the amorphous percentage of the composite membranes increased with the value of 71.8–77.7%. The crystallinity of the membrane increased with the addition of 6–10 wt.% SiO_2_, with the crystalline percentage of 24.9–28.2%, consequently causing the proton conductivity of the NMPC/PVA-SiO_2_ composite membranes with these compositions to decrease.

#### 3.3.2. Morphological Studies

The morphology of the modified composite membrane with SiO_2_ filler (NMPC/PVA-SiO_2_) was observed using FESEM and elemental mapping to observe the distribution of SiO_2_ filler in the polymer matrix. Figure 8 shows the FESEM micrographs of the cross-sectional views for the NMPC/PVA-SiO_2_ composite membranes with different compositions. All composite membranes were observed to be homogeneously mixed, and the structure of the composite membranes appeared rough and fibrous with noticeable pores (Figure 8a–g). However, no phase separation occurred when the polymer matrix and SiO_2_ filler were mixed, hence proving that all of the materials were compatible with each other when producing the NMPC/PVA-SiO_2_ composite membranes.

Appendix A shows the FESEM micrographs of the elemental mapping for all NMPC/PVA-SiO_2_ composite membranes with different SiO_2_ loadings in this study. The dispersion of inorganic filler in the polymer matrix had a large influence on the performance of the composite membranes. The structure of the SiO_2_ inorganic filler provided proton channels, and its dispersion affected the physical properties and conductivity of the composite membrane [20,65,66,67]. The uniform dispersion of SiO_2_ filler was very important for forming proton conduction channels, which consequently led to a high proton conductivity in the composite membrane. The SiO_2_ particles were dispersed uniformly, and no significant aggregation was observed in the FESEM images when 0.5–4 wt.% SiO_2_ was added into the polymer matrix (Appendix A). However, when the SiO_2_ loading was increased to values exceeding 6 wt.% (Appendix A), a uniform dispersion with a slight agglomeration of SiO_2_ particles was clearly visible in the polymer matrix. This tendency toward agglomeration at higher loadings was most likely due to the high and fine content of SiO_2_ particles, resulting in the occurrence of a high surface area with high surface energy, thus inducing increased agglomeration. The morphology observed through these FESEM images showed that when SiO_2_ particles were excessively added into the NMPC/PVA polymer matrix, they affected the performance of the composite membranes in regard to the proton conductivity, water uptake, and IEC.

#### 3.3.3. Thermal Stability

The thermal stability of the NMPC/PVA-50 composite membrane and NMPC/PVA-SiO_2_ composite membranes was determined through TGA analysis to ensure their performance in fuel cell applications. The TGA curve of the NMPC/PVA composite membrane in Figure 9 shows three stages of weight loss, similar to those discussed in Section 3.2.3. The TGA curves shown in Figure 9 show that all produced NMPC/PVA-SiO_2_ composite membranes (0.5–10 wt.%) followed degradation behavior that was very similar to that of the unmodified NMPC/PVA composite membrane. Figure 9 (I, II, and III) exhibited that there were three stages of degradation in the thermogram for all produced NMPC/PVA-SiO_2_ composite membranes. The first stage of degradation occurred at the range of 100–150 °C, with a weight loss of approximately 9–11% that was due to the loss of absorbed water content and weakly bound molecules in the membrane matrix. The second stage of degradation was due to the thermal degradation of the cross-linking network in the membrane, as well as the loss of PVA polymer and groups present in SiO_2_ in the range of 240–300 °C. The third stage of degradation occurred at temperatures of approximately 380–470 °C, which was caused by the thermal decomposition of the main polymer chain in the membrane matrix.

It can be observed that from the TGA curves of NMPC/PVA-65 composite membrane (Figure 4) and NMPC/PVA-SiO_2_ composite membranes (Figure 9), the main decomposition stage was very close or similar, which indicated that similar amount of component was decomposed as a function of temperature. However, the rate of decomposition might have changed after the heating and thermal degradation process. Based on the TGA curves and tables of thermal stability analysis (Appendix A), it can be seen that the onset or initial decomposition temperature has significantly increased from 80 °C in NMPC/PVA-65 composite membrane to approximately 100 °C in all of the NMPC/PVA-SiO_2_ composite membranes, while the residues remained very close or similar in both types of membranes. The initial temperature of the main degradation peak in the first stage shifted to a higher temperature (from 80 °C to 100 °C) when the SiO_2_ fillers were added to the membrane matrix, thereby showing increased thermal stability. Appendix A shows the thermal stability analysis of NMPC/PVA-50 composite membrane and NMPC/PVA-SiO_2_ composite membranes (0.5–10 wt.%) according to the three stages of degradation and weight loss (I, II, and III) as labeled in Figure 9. In addition, the NMPC/PVA-SiO_2_ (10 wt.%) composite membrane also exhibited a higher residue content or mass (approximately 30%) than the NMPC/PVA-50 composite membrane without SiO_2_ filler (17%) and other NMPC/PVA-SiO_2_ composite membranes with different compositions (20–28%). This increase proved that chemical modification by the addition of SiO_2_ filler into the membrane matrix could improve the thermal stability of the resulting composite membrane for use in fuel cell applications, remaining stable in the desired operating temperature range of 80–100 °C.

#### 3.3.4. Dynamic Mechanical Analysis (DMA)

A DMA analysis of the NMPC/PVA-SiO_2_ composite membranes was used to study the influence of different interactions between the membrane matrix and SiO_2_ filler on the mechanical properties of the composite membranes. Figure 10a–g shows the tan δ and storage modulus against temperature for the NMPC/PVA-SiO_2_ composite membranes with different compositions, while Appendix A shows the comparisons of tan δ curves for all NMPC/PVA-SiO_2_ composite membranes. Figure 10a–g shows that T_g_ occurred in the temperature range of 118–130 °C for all composite membranes. The tan δ curve also displayed that the range of T_g_ peak heights for all composite membranes was 0.5–0.7, indicating that all membranes were in the amorphous phase. Figure 10 exhibited that the NMPC/PVA-SiO_2_ (1 wt.%) composite membrane has the highest T_g_ peak value on the tan δ curve which was around 0.7 with increasing amorphous phase. Other than that, the NMPC/PVA-SiO_2_ composite membranes with SiO_2_ loadings of 6 wt.% and 10 wt.% showed lower T_g_ peaks, with values around 0.5, respectively, corresponding to the decrease of amorphous phase in the membrane, in which has been discussed on XRD analysis (Section 3.3.1). Based on all graphs (Figure 10a–g), another peak was likely to exist at temperatures below 35 °C, although the data displayed did not cover a sufficiently low temperature for full peak capture.

Besides that, Figure 10a–g also shows that two peaks appeared in all tan δ curves. The two or multiple tan δ peaks appeared in most polymers and the interpretation for this occurrence varies between the system of polymers. Different types or phases of polymers (crystalline, semi-crystalline, or amorphous) displayed different behavior or mechanisms due to temperature relaxations in the presented phase as well as the relative miscibility of the components in polymers. Moreover, the T_g_ and modulus values will be affected due to the heating rate, test frequency, and polymer structures, such as polymer chain rigidity, polymer chain flexibility, crystallization, and the degree of cross-linking [55,57].

Figure 10a–g also shows the storage modulus of all NMPC/PVA-SiO_2_ composite membranes, and the value of the storage modulus for all composite membranes decreased with an increasing temperature until it reached a temperature of approximately 100 °C. Moreover, the trends shown for each storage modulus were considerably inconsistent, which might be due to the thickness of membranes and undetermined degree of cross-linking. When the SiO_2_ filler was added into the membrane matrix with a loading range of 0.5–4 wt.%, the value of the storage modulus increased (around 4.80 × 10^5^–3.46 × 10^6^ Pa) and then experienced a slight decrease as the SiO_2_ content was further increased exceeding 4 wt.%. This storage modulus value then increased slightly and continued to decrease (5.00 × 10^5^ Pa) when SiO_2_ loading of 10 wt.% was added into the membrane matrix. Despite having a slight decrease, the initial storage modulus value (at a temperature of approximately 30 °C) could be maintained around 100 °C in the NMPC/PVA-SiO_2_ (10 wt.%) composite membrane (Figure 10g), demonstrating the mechanical stability of the membrane and its potential use within the optimum range of low operating temperature (80–100 °C) of fuel cell applications.

#### 3.3.5. Water Uptake, Swelling Ratio, Ion Exchange Capacity, and Proton Conductivity

Effective proton transfer in the PEM depended on water molecules that provided proton transport and formed a network of hydrogen bonds in the membrane matrix. Therefore, a high water content was a major factor in obtaining a high proton conductivity. The water uptake, swelling area, and swelling thickness of the composite membrane were measured at room temperature and are listed in Table 2. The water uptake increased with an increasing SiO_2_ loading in the membrane matrix (45.7–55.7%) and then experienced a slight decrease to a certain extent. The presence of fillers having hydrophilic groups (Si-OH) in SiO_2_ that absorbed or attracted water was one of the factors that contributed to the increase in water uptake, as well as the uniform dispersion of fillers in the membrane matrix. The water uptake value obtained by the NMPC/PVA-SiO_2_ composite membrane was higher than that of the NMPC/PVA composite membranes without SiO_2_. The highest measured water uptake was 55.7% for the NMPC/PVA-SiO_2_ composite membrane (4 wt.%). When the SiO_2_ loading was further increased to 8 wt.%, the water uptake decreased (46.5%), possibly due to the addition of excess filler, which reduced the size of the ion channels that served as the main water storage space in the membrane matrix [66]. The agglomeration of filler at high SiO_2_ loading concentrations also resulted in a reduction of number of polar groups (Si-OH) presented for the purpose of water uptake in the membrane [68].

Table 2 also exhibits the values of the swelling ratio, which were the swelling area and swelling thickness of all NMPC/PVA-SiO_2_ composite membranes. Generally, it is known that a higher water uptake can lead to a higher swelling ratio. Table 2 shows that both the swelling area and swelling thickness and water uptake show similar trends. The swelling ratio increased with an increasing water uptake value and decreased slightly as the water uptake percentage decreased. All NMPC/PVA-SiO_2_ composite membranes underwent a cross-linking process to avoid dissolution and to reduce excessive swelling of the composite membrane. The NMPC/PVA-SiO_2_ composite membrane (4 wt.%) showed the highest swelling ratio, which was correlated with the highest water uptake value than the NMPC/PVA-50 composite membrane. However, this membrane still demonstrated good stability when compared to the unmodified NMPC membrane. Nevertheless, there was no clear trend regarding the increase or decrease of the swelling ratios.

The IEC depended on the number of functional groups present in the composite membrane, which was also an indicator of the number of active groups present for each mass of material to facilitate ion transfer and proton conduction. Table 2 shows the IEC values of the NMPC/PVA-SiO_2_ composite membranes with different compositions. The increase in SiO_2_ loadings (0.5–4 wt.%) in the NMPC/PVA-SiO_2_ composite membranes increased the IEC magnitude from 0.43 to 0.56 mequiv g^−1^ and decreased as the SiO_2_ loadings were further increased. This increase in the IEC value was likely due to the active groups (Si-OH) in the filler that was mixed with the NMPC/PVA membrane matrix. The IEC trend is the same as the water uptake and swelling ratio, therefore when the water uptake and swelling ratio decreased, the IEC value also decreased. After the IEC value for NMPC/PVA-SiO_2_ composite membrane (6–8 wt.%) decreased, there was a slight increase in the IEC value for NMPC/PVA-SiO_2_ composite membrane (10 wt.%), which was due to the aggregation of filler material. However, the aggregation was likely to be more uniform compared to the presence of SiO_2_ filler with loadings of 6–8 wt.%. This inconsistent trend might also be contributed by different degrees of cross-linking of composite membranes.

The fuel cell performance efficiency depended on the proton conductivity, which was also an important feature of a fuel cell membrane. The presence of water was very important for proton conduction in the membrane, thus proving that water uptake was closely related to the improvement in proton conductivity. The proton conductivity of commercial Nafion 212 membrane was measured experimentally under hydrated conditions. The proton conductivity of Nafion 212 has increased from 1.25 × 10^−2^ S cm^−1^ (at 25 °C) to 2.06 × 10^−2^ S cm^−1^ (at 80 °C), and then decreased when the temperature reached 100 °C due to membrane deterioration. As expected, the proton conductivities of Nafion 212 are higher by two orders of magnitude compared to the NMPC/PVA and NMPC/PVA-SiO_2_ membranes in the study. The proton conductivity of the NMPC/PVA-SiO_2_ composite membranes with different compositions was measured from 25–100 °C under hydrated conditions and the results are shown in Table 2. Regarding the NMPC/PVA-SiO_2_ composite membranes, it was observed that the proton conductivity increased with an increasing SiO_2_ loadings in the membrane matrix with a loading range of 0.5–4 wt.% (3.20–5.08 × 10^−4^ S cm^−1^ at 100 °C). The proton conductivity increased with an increasing temperature due to the increase in flexibility of the polymer chain and the mobility of water molecules at high temperature. Moreover, the presence of the -OH groups on SiO_2_ might also provide additional conduction sites for proton transfer. The proton conductivity decreased when SiO_2_ was added up to 8 wt.% (4.38 × 10^−4^ S cm^−1^ at 100 °C) and then increased slightly when 10 wt.% SiO_2_ was added into the membrane matrix.

The changes in proton conductivity were due to several reasons, namely, when the SiO_2_ loading was less than 4 wt.%, the fine and uniform dispersion of SiO_2_ in the mem-brane matrix promoted a decrease in crystallinity and caused the amorphous phase con-tent to increase. This amorphous phase changes have promoted proton transport through the membrane. Moreover, the interaction of hydrogen bonds between the membrane matrix and SiO_2_ could also induce the formation of continuous proton transport channels in the membrane. In addition, the increase in water uptake was beneficial for proton mobility and hydrogen bonding network formation. However, when the SiO_2_ loading reached 6–10 wt.%, as seen in the FESEM mapping image (Appendix A), the agglomeration of SiO_2_ in the membrane matrix could result in proton conduction pathways with high tortuosity, thus decreasing the proton conductivity to a certain extent [69]. Therefore, based on the results shown in Table 2, the NMPC/PVA-SiO_2_ (4 wt.%) composite membrane exhibited the highest proton conductivity, which was 5.08 × 10^−4^ S cm^−1^ at 100 °C. This increase in proton conductivity was better than that of the NMPC/PVA-50 composite membranes without SiO_2_ (2.22 × 10^−4^ S cm^−1^ at 100 °C), thereby supporting that the improvement in conductivity performance was due to the presence of the hygroscopic filler SiO_2_. However, in this study, the NMPC/PVA-SiO_2_ (4 wt.%) composite membrane was not used to proceed for fuel cell polarization testing because of its low conductivity magnitude (10^−4^ S cm^−1^). The fabrication of MEA with this NMPC/PVA-SiO_2_ membrane will further increase the ohmic losses, consequently, leading to mass transport losses and performance losses of a fuel cell.

All membrane samples exhibited a positive temperature-dependent proton conductivity, indicating an activated thermal process. The regression values obtained from the linear line were approximately R^2^ ≈ 1, and the proton conductivity increased with an increasing temperature for all composite membranes, which can be expressed via Arrhenius plots using the following equation [70]:(6)$$\ln\sigma =-\;\frac{1000E_{a}}{\mathrm{RT}}$$
where σ is the proton conductivity, R is the universal gas constant (8.314 J mol^−1^ K^−1^), E_a_ is the activation energy (kJ/mol), and T refers to the absolute temperature (K).

The activation energy or minimum energy required for proton conduction over the membrane could be obtained from the slope of the linear line (Appendix A). The NMPC/PVA-SiO_2_ (4 wt.%) composite membrane showed an activation energy of 11.87 kJ/mol, which was lower than the activation energy of the NMPC/PVA-50 composite membrane without SiO_2_ (12.09 kJ/mol). The achieved activation energy could be attributed to the Grotthuss mechanism, which includes values of less than 15 kJ/mol [71]. The above activation energy value was also supported when the addition of SiO_2_ induced the generation of a continuous proton conduction network for rapid proton transportation with low energy resistance through the Grotthuss mechanism. Furthermore, the linear lines obtained from the Arrhenius plots for the conductivity values indicated that the proton conduction mechanism was primarily determined by Grotthuss or hopping conductor species. The deviation of several points from the linear plots in the composite membranes suggested that the vehicle mechanism might also contribute to proton conduction [72].

In addition, Table 3 exhibited the comparisons of Nafion-based membranes, PVA-based membranes, and chitosan-based membranes in term of their water uptake, IEC and proton conductivity properties. Based on the Table 3, it can be observed that PVA-based and chitosan-based membranes possessed lower proton conductivity than Nafion-based membranes. However, the proton conductivities for both PVA-based and chitosan-based membranes were comparable. Even though the unmodified NMPC membrane that has been studied in this work showed the lowest proton conductivity, however, the conductivity could be enhanced with further modifications on the NMPC membrane and led to the improvement of one or two magnitudes. Other than that, the NMPC/PVA-SiO_2_ composite membrane seemed to have better proton conductivity than most of the chitosan-based and PVA-based membranes listed in the table or comparable to them though cannot be compared to the Nafion-based membranes.

## 4. Conclusions

In summary, this work proved that chitosan could be successfully modified through the phosphorylation process, which introduced phosphonic acid groups to convert chitosan into *N*-methylene phosphonic chitosan (NMPC). This modification produced a water-soluble functionalized polymer that had a high degree of hydrophilicity compared to pristine chitosan, which is insoluble in water. The effect of different NMPC biopolymer contents in the NMPC/PVA composite membranes was thoroughly studied in this work. The NMPC/PVA composite membranes were prepared by varying the composition of NMPC (30–70%) in the membrane, and physical and chemical characterizations were carried out. Based on the results obtained from the analyses, it could be concluded that the NMPC/PVA composite membrane with an NMPC content of 50 wt.% showed the best performance among the other produced membranes in regard to the proton conductivity, water uptake, and IEC (8.76 × 10^−5^ S cm^−1^, 51.9%, and 0.45 mequiv g^−1^, respectively). In addition, this membrane could be further improved by the addition of hygroscopic filler into the membrane matrix. As the NMPC/PVA-50 composite membrane showed the best performance, this membrane was chosen as the base membrane to be modified with the addition of different loadings (0.5–10 wt.%) of inorganic filler (SiO_2_). The results were expected to improve, and it was proven that the NMPC/PVA-SiO_2_ (4 wt.%) composite membrane exhibited the highest proton conductivity of 5.08 × 10^−4^ S cm^−1^ at 100 °C with an IEC value of 0.56 mequiv g^−1^. The NMPC/PVA-SiO_2_ composite membrane displayed better performance than the unmodified NMPC/PVA membrane. However, the NMPC/PVA-based membranes were not comparable to the commercial Nafion membrane. Nevertheless, this chitosan-based membranes could be potentially used as PEM due to few advantages and properties including low-cost, biodegradable and could be used in other low power applications. Hence, further research will be conducted in the future, by replacing the inorganic filler with ionic liquids, which is expected to increase the conductivity values. In conclusion, the NMPC/PVA-based composite membrane needs extensive enhancement, especially on the conductivity value before the membrane could potentially be used as a PEM in fuel cell applications.

## Figures and Tables

**Figure 1 membranes-11-00675-f001:**
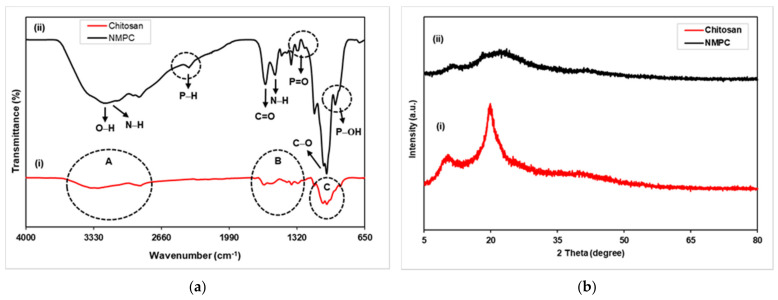
(**a**) FTIR spectra of chitosan and *N*-methylene phosphonic chitosan (NMPC) and (**b**) XRD diffractograms of chitosan and NMPC.

**Figure 2 membranes-11-00675-f002:**
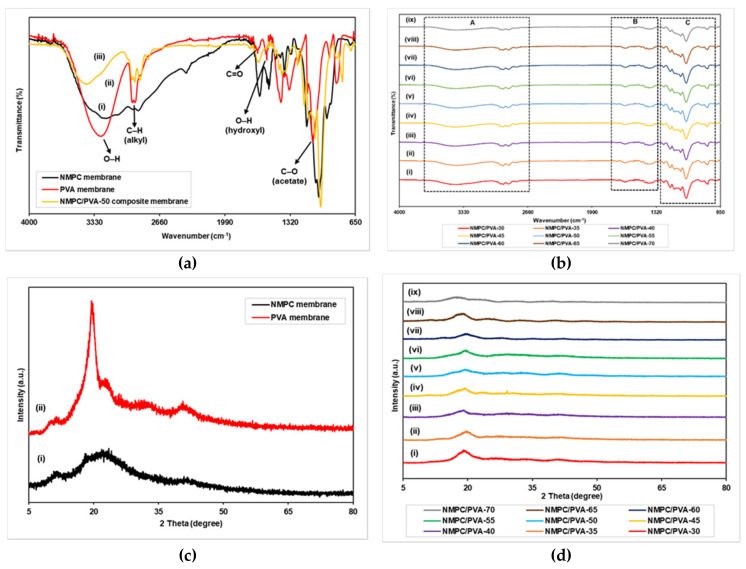
(**a**) FTIR spectra of NMPC membrane, PVA membrane, and NMPC/PVA-50 composite membrane, (**b**) FTIR spectra of NMPC/PVA composite membranes with different compositions, (**c**) XRD spectra of NMPC and PVA membrane, and (**d**) XRD spectra of NMPC/PVA composite membranes with different compositions.

**Figure 3 membranes-11-00675-f003:**
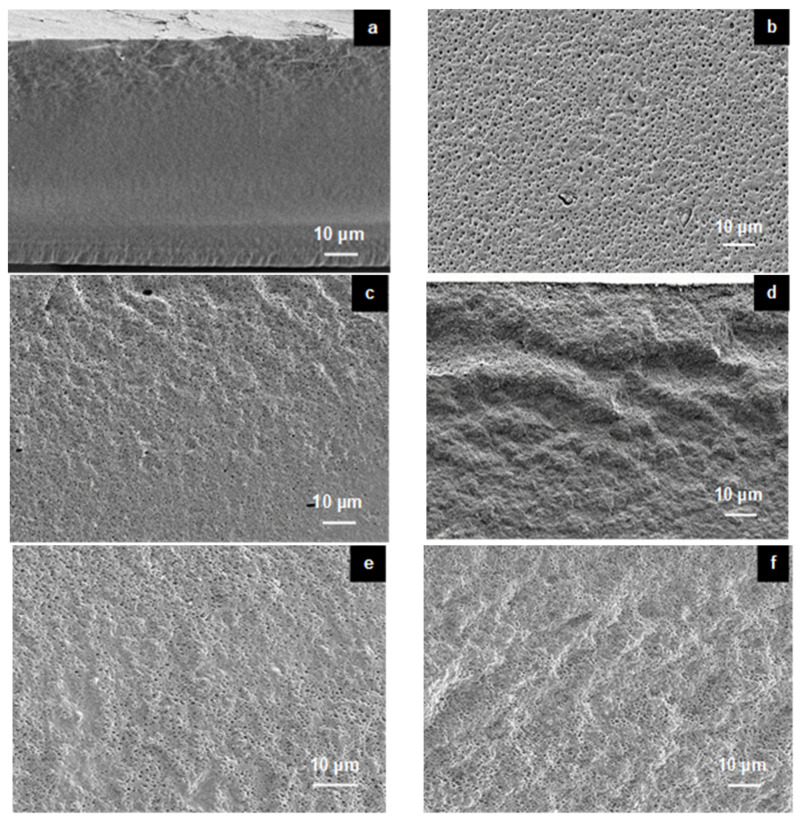
FESEM micrographs of the cross-sectional views for the (**a**) NMPC membrane, (**b**) PVA membrane, (**c**) NMPC/PVA-30 membrane, (**d**) NMPC/PVA-35 membrane, (**e**) NMPC/PVA-40 membrane, (**f**) NMPC/PVA-45 membrane, (**g**) NMPC/PVA-50 membrane, (**h**) NMPC/PVA-55 membrane, (**i**) NMPC/PVA-60 membrane, (**j**) NMPC/PVA-65 membrane, and (**k**) NMPC/PVA-70 membrane.

**Figure 4 membranes-11-00675-f004:**
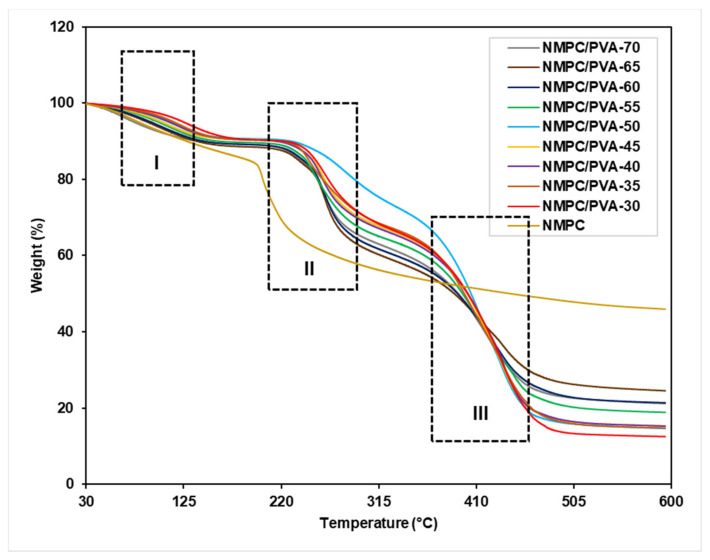
TGA curves of the NMPC membrane and NMPC/PVA composite membranes with different compositions.

**Figure 5 membranes-11-00675-f005:**
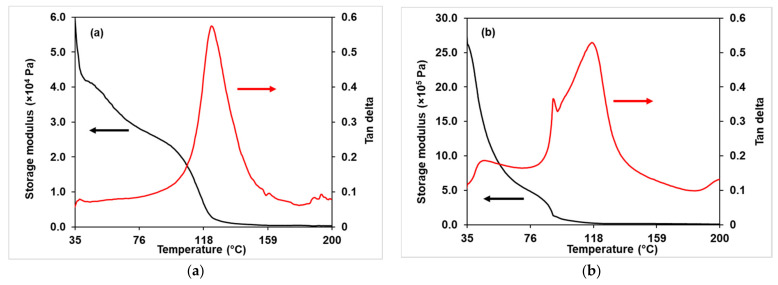
Tan δ and storage modulus of (**a**) NMPC/PVA-30 membrane, (**b**) NMPC/PVA-35 membrane, (**c**) NMPC/PVA-40 membrane, (**d**) NMPC/PVA-45 membrane, (**e**) NMPC/PVA-50 membrane, (**f**) NMPC/PVA-55 membrane, (**g**) NMPC/PVA-60 membrane, (**h**) NMPC/PVA-65 membrane, and (**i**) NMPC/PVA-70 membrane.

**Figure 6 membranes-11-00675-f006:**
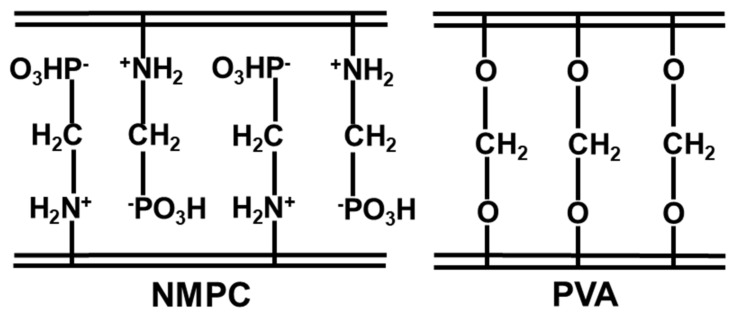
Schematic of zwitterionic structure in NMPC/PVA composite membrane.

**Figure 7 membranes-11-00675-f007:**
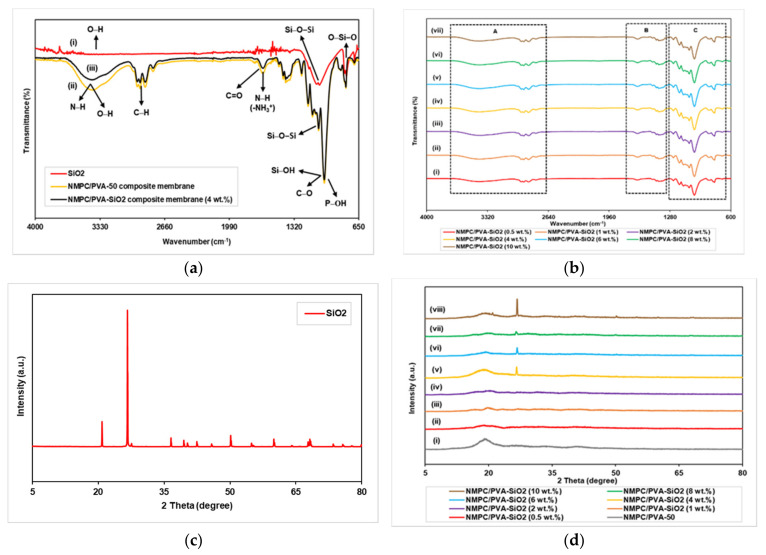
(**a**) FTIR spectra of SiO_2_ filler, NMPC/PVA-50 and NMPC/PVA-SiO_2_ (4 wt.%) composite membrane, (**b**) FTIR spectra of NMPC/PVA-SiO_2_ composite membrane with different compositions, (**c**) XRD spectrum of commercial SiO_2_ filler obtained experimentally, and (**d**) XRD spectra of NMPC/PVA and NMPC/PVA-SiO_2_ composite membrane with different compositions.

**Figure 8 membranes-11-00675-f008:**
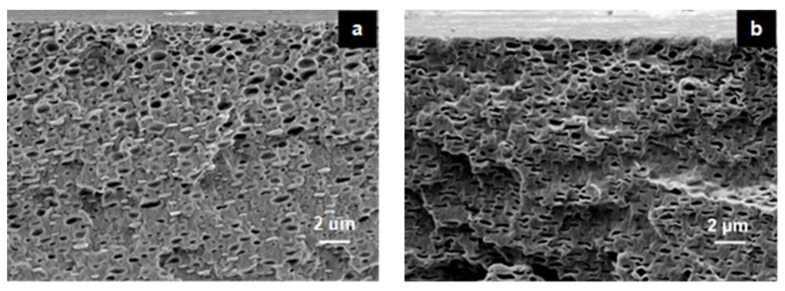
FESEM micrographs of the cross-sectional views of the (**a**) NMPC/PVA-SiO_2_ (0.5 wt.%), (**b**) NMPC/PVA-SiO_2_ (1 wt.%), (**c**) NMPC/PVA-SiO_2_ (2 wt.%), (**d**) NMPC/PVA-SiO_2_ (4 wt.%), (**e**) NMPC/PVA-SiO_2_ (6 wt.%), (**f**) NMPC/PVA-SiO_2_ (8 wt.%), and (**g**) NMPC/PVA-SiO_2_ (10 wt.%) composite membranes.

**Figure 9 membranes-11-00675-f009:**
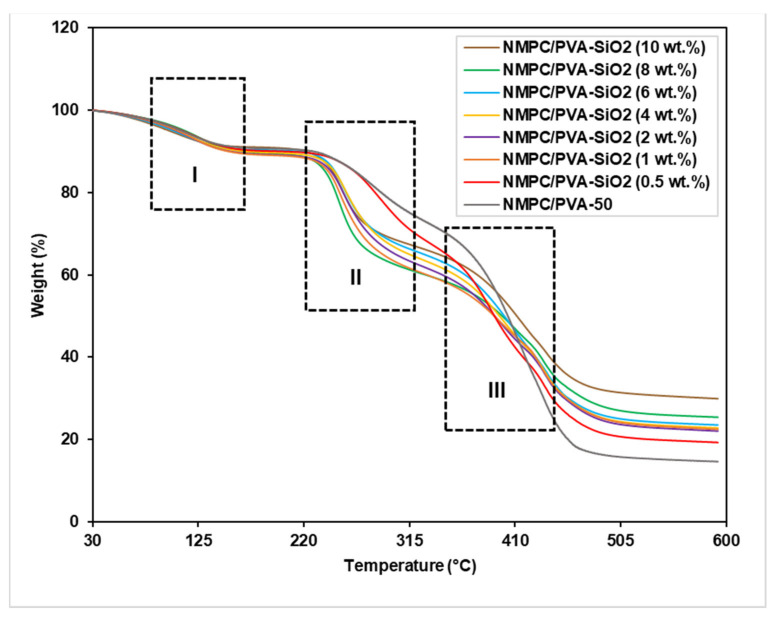
TGA curves of the NMPC/PVA-50 composite membrane and NMPC/PVA-SiO_2_ composite membranes with different compositions.

**Figure 10 membranes-11-00675-f010:**
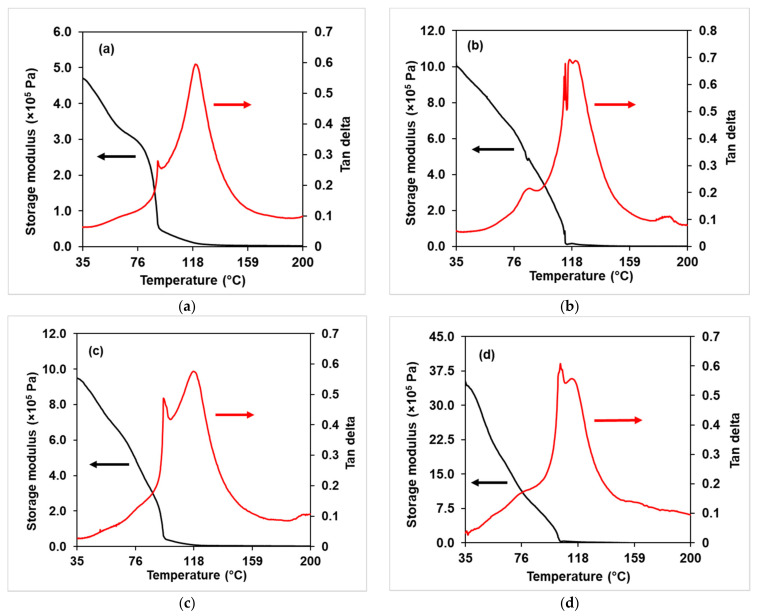
Tan δ and storage modulus of the (**a**) NMPC/PVA-SiO_2_ (0.5 wt.%), (**b**) NMPC/PVA-SiO_2_ (1 wt.%), (**c**) NMPC/PVA-SiO_2_ (2 wt.%), (**d**) NMPC/PVA-SiO_2_ (4 wt.%), (**e**) NMPC/PVA-SiO_2_ (6 wt.%), (**f**) NMPC/PVA-SiO_2_ (8 wt.%), and (**g**) NMPC/PVA-SiO_2_ (10 wt.%) composite membrane.

**Table 1 membranes-11-00675-t001:** Thickness, water uptake, swelling area, swelling thickness, IEC, and proton conductivity of the NMPC/PVA composite membranes with different compositions.

Membrane Sample	Thickness (mm)	Water Uptake (%)	Swelling Area (%)	Swelling Thickness (%)	IEC (mequiv g^−1^)	Proton Conductivity (10^−5^ S cm^−1^)
NMPC/PVA-30	0.06 ± 0.02	32.1 ± 5.2	15.1 ± 3.6	46.5 ± 5.9	0.24 ± 0.08	2.61 ± 0.29
NMPC/PVA-35	0.07 ± 0.03	37.3 ± 5.1	25.4 ± 7.1	51.2 ± 5.3	0.39 ± 0.02	3.45 ± 0.38
NMPC/PVA-40	0.08 ± 0.01	44.4 ± 2.3	29.6 ± 16.3	58.6 ± 5.9	0.40 ± 0.03	4.96 ± 0.33
NMPC/PVA-45	0.08 ± 0.02	48.4 ± 3.5	33.4 ± 7.1	61.3 ± 1.6	0.42 ± 0.06	6.59 ± 0.19
NMPC/PVA-50	0.09 ± 0.01	51.9 ± 2.3	35.1 ± 16.3	62.2 ± 3.3	0.45 ± 0.02	8.76 ± 0.16
NMPC/PVA-55	0.10 ± 0.02	49.5 ± 5.5	33.3 ± 12.9	54.3 ± 2.3	0.38 ± 0.04	7.47 ± 0.25
NMPC/PVA-60	0.10 ± 0.04	48.3 ± 6.5	30.7 ± 7.8	51.2 ± 2.8	0.36 ± 0.03	5.38 ± 0.41
NMPC/PVA-65	0.11 ± 0.01	42.5 ± 6.2	24.1 ± 5.1	45.1 ± 2.2	0.33 ± 0.02	4.26 ± 0.34
NMPC/PVA-70	0.12 ± 0.03	39.1 ± 2.9	17.8 ± 9.8	39.2 ± 1.9	0.29 ± 0.07	3.59 ± 0.46

**Table 2 membranes-11-00675-t002:** Thickness, water uptake, swelling area, swelling thickness, IEC, and proton conductivity at different temperature (25–100 °C) of the NMPC/PVA-50 and NMPC/PVA-SiO_2_ composite membranes with different compositions.

Membrane Sample	Thickness (mm)	Water Uptake (%)	Swelling Area (%)	Swelling Thickness (%)	IEC (mequiv g^−1^)	Proton Conductivity (10^−4^ S cm^−1^)				
						25 °C	40 °C	60 °C	80 °C	100 °C
NMPC/PVA-50	0.09 ± 0.01	51.9 ± 2.3	35.1 ± 16.3	62.2 ± 3.3	0.45 ± 0.02	0.87 ± 0.02	1.10 ± 0.03	1.69 ± 0.01	2.03 ± 0.02	2.22 ± 0.01
NMPC/PVA-SiO_2_ (0.5 wt.%)	0.10 ± 0.01	45.7 ± 2.0	28.4 ± 5.9	44.6 ± 8.8	0.43 ± 0.02	1.14 ± 0.04	1.78 ± 0.08	2.22 ± 0.01	2.87 ± 0.01	3.20 ± 0.06
NMPC/PVA-SiO_2_ (1 wt.%)	0.10 ± 0.02	47.1 ± 1.8	33.1 ± 10.8	48.3 ± 4.9	0.45 ± 0.02	1.36 ± 0.07	1.93 ± 0.09	2.47 ± 0.10	3.25 ± 0.02	3.81 ± 0.05
NMPC/PVA-SiO_2_ (2 wt.%)	0.10 ± 0.01	47.4 ± 2.8	37.6 ± 11.3	52.9 ± 3.6	0.49 ± 0.03	1.63 ± 0.06	2.28 ± 0.04	3.42 ± 0.14	3.94 ± 0.13	4.41 ± 0.14
NMPC/PVA-SiO_2_ (4 wt.%)	0.11 ± 0.01	55.7 ± 1.9	43.9 ± 5.5	63.1 ± 1.0	0.56 ± 0.02	1.90 ± 0.09	2.38 ± 0.09	3.51 ± 0.07	4.65 ± 0.06	5.08 ± 0.05
NMPC/PVA-SiO_2_ (6 wt.%)	0.11 ± 0.02	50.6 ± 1.9	35.3 ± 6.6	55.8 ± 4.8	0.44 ± 0.03	1.82 ± 0.07	2.22 ± 0.17	3.42 ± 0.11	4.16 ± 0.10	4.67 ± 0.09
NMPC/PVA-SiO_2_ (8 wt.%)	0.12 ± 0.03	46.5 ± 1.2	27.6 ± 6.1	51.4 ± 5.1	0.37 ± 0.03	1.56 ± 0.08	1.96 ± 0.09	2.79 ± 0.10	3.53 ± 0.05	4.38 ± 0.08
NMPC/PVA-SiO_2_ (10 wt.%)	0.13 ± 0.02	48.8 ± 1.2	32.8 ± 6.1	54.1 ± 8.2	0.41 ± 0.02	1.71 ± 0.16	2.22 ± 0.08	3.05 ± 0.09	3.97 ± 0.08	4.53 ± 0.05
Nafion 212	0.02 ± 0.01	-	-	-	-	* 1.25 ± 0.03	* 1.58 ± 0.05	* 1.80 ± 0.04	* 2.06 ± 0.03	* 1.87 ± 0.06

* data with the unit of 10^−2^ S cm^−1^

**Table 3 membranes-11-00675-t003:** Comparisons of water uptake, IEC, and proton conductivity for Nafion-based membranes, PVA-based membranes, and chitosan-based membranes.

Membrane	Water Uptake (%)	IEC (mequiv g^−1^)	Proton Conductivity (S cm^−1^)	Applications	References
PVdF-coHFP/Nafion	33.8	-	1.00 × 10^−3^	DMFC	[73]
Nafion-sulfonated PVdF coated	13.0	-	5.91 × 10^−3^	DMFC	[74]
CS/PVS-Nafion	29.1	-	7.01 × 10^−2^	DMFC	[75]
Nafion/CNT	29.5	0.90	7.35 × 10^−2^	DMFC	[76]
Nafion 212	-	-	2.06 × 10^−2^	PEM	This study
SPVA-SPTA	150.4	0.45	8.80 × 10^−4^	PEM	[77]
PVA-CS-CNC	78	-	6.42 × 10^−4^	DMFC	[78]
CS/SPVA-SSA	220	2.60	2.58 × 10^−4^	PEM	[77]
Ph/CS-NH_4_SCN	-	-	2.42 × 10^−5^	PEM	[79]
NMPC	-	-	2.74 × 10^−6^	PEM	This study
NMPC-OMPk	-	-	1.43 × 10^−5^	PEM	[41]
NMPC/PVA	51.9	0.45	2.22 × 10^−4^	PEM	This study
NMPC/PVA-SiO_2_	55.7	0.56	5.08 × 10^−4^	PEM	This study

## Data Availability

Not applicable.

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
