# Peer review of "Hybrid Composite Membrane of Phosphorylated Chitosan/Poly (Vinyl Alcohol)/Silica as a Proton Exchange Membrane"

_membranes, 2021, doi:10.3390/membranes11090675_

Round 1

Reviewer 1 Report

Reviewer comments-Membranes

 Hybrid Composite Membrane of Phosphorylated Chitosan/Poly(Vinyl Alcohol)/Silica as a Proton Exchange Membrane

In this study many experiments are reported but it is not enough for OPEMFC.

First of all, the proton conductivity values are very weak that are not acceptable for fule cell performance.

Nowadays, almost all manuscript for fuel cell have Fuel Cell polarization curve and I think this work also needs these curves.

The comparison with other Chitosan based membranes should be reported as a table.

One of the most problem of Chitosan based membrane is durability. I cannot see any test for chemical durability.

SEM EDX mapping should be presented to see the SiO2 distributions in the membranes.

The reason for choosing 0.5 to 10% SiO2 should be clarified.

In total, this manuscript has lots of characterizations but not important ones for PEMFC.

Author Response

Editor-in-chief,

Membrane MDPI 

Manuscript ID: membranes-1268211

Title: Hybrid Composite Membrane of Phosphorylated Chitosan/Poly (Vinyl Alcohol)/Silica as a Potential Proton Exchange Membrane                                                                            

Dear Editor,

Thank you for your response and reviewers’ useful comments on our manuscript. We have modified the manuscript in response to the extensive and insightful comments. The detailed corrections and the comments are made point by point which is given below (as attached).

Reviewer 2 Report

The manuscript describes the manufacturing and characterization of a specific polymer membrane for
the use in proton exchange fuel cells. The paper written in a clear and comprehensive style.

Suggestions: 
In the introduction the work is motivated to deliver a substitute for the Nafion type membranes.
Alternative membranes from the literature are discussed -> and the parameters 'max. power density' and 'proton conductivity'
are used  to characterize them.
The 'max. power density' and 'proton conductivity'' should also be given for a typical Nafion membrane to show the impact.
For this parameter the type of fuel cell, i.e. direct methanol or hydrogen/ air fuel cell, should be 
discussed separately. These two types have very differnt power densities and different needs for the membrane,
e.g. methanol cross over.
The fact that commercial Nafion membranes show much better performance is mentioned in the conclusion - which is
true for high power density applications. The authors could also point into other directions, where the chitosan based 
membranes would be of advantage becasue of low cost and bio-degradable, e.g. small fuel cells in biological environment or
other low power applications.

Figures 1,2,7 should show numbers at the y-axis - even if the unit is % or a.u.

Figure 1 used (a) and (b) to distinguish left and right picture.
In the sub figures (a) and (b) is used again to distinguish the red and black line -> this is redundant and should be removed.
See also Fig. 2, 7.

Author Response

Editor-in-chief,

Membrane MDPI

Manuscript ID: membranes-1268211

Title: Hybrid Composite Membrane of Phosphorylated Chitosan/Poly (Vinyl Alcohol)/Silica as a Potential Proton Exchange Membrane

Dear Editor,

Thank you for your response and reviewers’ useful comments on our manuscript. We have modified the manuscript in response to the extensive and insightful comments. The detailed corrections and the comments are made point by point which is given below. (as attached).

Reviewer 3 Report

The authors present a hybrid composite membrane for application as a Proton Exchange Membrane, for example to be used in fuel cells. Similar Membranes have been shown before and conductivity remains, even for the optimized membrane, significantly below current commercial membranes. Therefore the paper will be more relevant to readers looking for an understanding of how membrane properties can be tuned in general, and not relevant for readers looking for alternative membranes for theis application.

The membranes are carefully charcterized by a number of complementary analytical methods and the presentation of the results is generally clear and easy to follow. 

The only exception are the results of the DMA, which show, as the authors note, no consistency. I would assume that the degree of crosslinking, which has not been determined, might be responsible for part of this. While it will not be possible to determine the crosslinking now, this should at least be mentioned as a possible cause.

In my opininon the manuscript should be published with the following minor revisions:

1) Mention degree of crosslinking as a possible cause for mechanical properties and possibly also for maximum in IEC and conductivity.

2) Discuss why TGA of silica-filled membranes is similar to NMPC/PVA-65.

3) References are generally well chosen, except for the references in the first paragraph of the introduction. Here 2-3 recent reviews on polymer electrolyte membranes should be cited. The choice of reviews in this field is very big.

4) Line 218: Mention type and wavelength of radiation used

5) Line 361 and 363: Where do C=O and acetate groups come from in Polyvinylalcohol?

6) Figure 2a: Mention which composite membrane is shown

7) Line 493: If glass temperature is taken as the maximum of tanδ then it is closer to 120°C.

8) Line 599 ff SiO2 are crystals, not molecules

Author Response

(The authors gave the same response as above.)

Reviewer 4 Report

The manuscript reported a novel proton exchange membrane based on chitosan as one of the natural biopolymers to replace Nafion membranes in fuel cell applications. By modifying chitosan through polymer blending and incorporation of inorganic filler, a series of NMPC/PVA-SiO2 composite membranes are obtained. The thermal stability, chemical structure, mechanical stability, morphology, and the most relevant physicochemical properties to fuel cell applications have been investigated and discussed.

I consider the content of this manuscript will definitely meet the reading interests of the readers of Membranes journals. Although the proton conductivity of the final membrane material still needs to be improved in this study in comparison to commercial Nafion membranes, the overall characterization of this manuscript is still comprehensive and clear. Therefore, I suggest giving a minor revision and accepting it after clarifying some issues.

The details of my comments can be found in a separate word file.

Author Response

(The authors gave the same response as above.)

Round 2

Reviewer 1 Report

The manuscript has been well revised.